# Defining a Haplotype Encompassing the *LCORL-NCAPG* Locus Associated with Increased Lean Growth in Beef Cattle

**DOI:** 10.3390/genes15050576

**Published:** 2024-04-30

**Authors:** Leif E. Majeres, Anna C. Dilger, Daniel W. Shike, Joshua C. McCann, Jonathan E. Beever

**Affiliations:** 1UTIA Genomics Center for the Advancement of Agriculture, Institute of Agriculture, University of Tennessee, Knoxville, TN 37996, USA; lmajeres@vols.utk.edu; 2Department of Animal Sciences, University of Illinois at Urbana-Champaign, Urbana, IL 61801, USA; adilger2@illinois.edu (A.C.D.); dshike@illinois.edu (D.W.S.); jcmccan2@illinois.edu (J.C.M.)

**Keywords:** *Bos taurus*, *LCORL*, *NCAPG*, whole-genome sequencing, body weight, haplotype

## Abstract

Numerous studies have shown genetic variation at the *LCORL-NCAPG* locus is strongly associated with growth traits in beef cattle. However, a causative molecular variant has yet to be identified. To define all possible candidate variants, 34 Charolais-sired calves were whole-genome sequenced, including 17 homozygous for a long-range haplotype associated with increased growth (*QQ*) and 17 homozygous for potential ancestral haplotypes for this region (*qq*). The *Q* haplotype was refined to an 814 kb region between chr6:37,199,897–38,014,080 and contained 218 variants not found in *qq* individuals. These variants include an insertion in an intron of *NCAPG*, a previously documented mutation in *NCAPG* (rs109570900), two coding sequence mutations in *LCORL* (rs109696064 and rs384548488), and 15 variants located within ATAC peaks that were predicted to affect transcription factor binding. Notably, rs384548488 is a frameshift variant likely resulting in loss of function for long isoforms of *LCORL*. To test the association of the coding sequence variants of *LCORL* with phenotype, 405 cattle from five populations were genotyped. The two variants were in complete linkage disequilibrium. Statistical analysis of the three populations that contained *QQ* animals revealed significant (*p* < 0.05) associations with genotype and birth weight, live weight, carcass weight, hip height, and average daily gain. These findings affirm the link between this locus and growth in beef cattle and describe DNA variants that define the haplotype. However, further studies will be required to define the true causative mutation.

## 1. Introduction

Body size is a trait frequently subject to selection in domesticated animals. This trait is of particular importance in species raised for meat as body size directly correlates to the size of the carcass and thus quantity of meat produced per animal. In beef cattle, body size can generally be described as a highly complex polygenic trait, with even the most impactful DNA variants explaining only a modest fraction of the variance observed [1]. However, there are a select few loci that can be considered highly impactful for body size, one of them being the *LCORL-NCAPG* locus on bovine chromosome 6. Over the years, many studies have indicated this locus as being associated with stature [1,2,3,4], birth weight [5,6,7,8], and carcass weight [9,10,11,12], as well as other carcass characteristics such as increased ribeye area and reduced adiposity [13], suggesting a potential role in increasing lean growth. These findings are supported by research in other species, including humans [14,15,16], dogs [17,18], horses [19,20,21], and sheep [22,23], which has also found variants surrounding this locus to be associated with increased body size.

However, while there is no doubt that this locus has a significant impact on body size, the biological mechanism mediating this effect remains unclear. The large effect size of this locus resulted in rapid selective sweeps in cattle [1,10,24,25] and other domesticated species [26,27,28]; the ensuing linkage disequilibrium (LD) has long confounded attempts to discern a causative mutation within the selected haplotype. Indeed, it has been shown the same haplotype at this locus is almost fixed in some breeds of cattle such as Brown Swiss and Montbeliard [1]. That same study found that in other breeds such as Charolais, this haplotype is present and very abundant, but not to the exclusion of other possible haplotypes for the region. This locus has been narrowed to a range as small as 591 kb [9], but further refinement remains complicated by the aforementioned LD.

Roughly half of this range encompasses a gene-dense region containing four genes: *LCORL*, *NCAPG*, *DCAF16*, and *FAM184B*. The other portion of this range is intergenic space upstream of *LCORL*. Many genome-wide association studies have found the most strongly associated markers with growth to be associated with the intergenic region upstream of *LCORL* [8,12,29,30], suggesting a mutation affecting the regulation of *LCORL* transcription may be responsible for the change in phenotype. Earlier studies pointed to a missense mutation in *NCAPG* [2,31], *NCAPGc.1326T>G*, but subsequent studies have found this mutation present in *q* haplotypes as well, implying this SNP is simply in high LD with the causative mutation in certain populations [29,32].

Recent evidence in other species places more weight in favor of *LCORL* as the gene responsible for this locus’ influence on body size and development. In dogs, a mutation that results in the loss of function of the long isoform of *LCORL* has been found to be exclusive to medium- and large-sized breeds of dogs [18]. A similar mutation exists in large-sized Pakistani goat breeds [26]; however, functional validation of these loss-of-function mutations has yet to be performed in either species. The long isoform of *LCORL* encodes for a protein dubbed PALI2 by Conway et al. [33]. This protein has not been directly characterized as of the time of writing, but it is postulated to have similar activity to PALI1, a protein encoded by *LCOR*, the paralog of *LCORL*.

PALI1 acts as an accessory protein to polycomb repressive complex 2 (PRC2), an essential protein complex that is responsible for the creation and maintenance of cellular identity by mono-, di-, and trimethylation of H3K27. This methylation represses transcription and is maintained from early development onward [34]. Disturbances of PRC2 function have been shown to cause abnormalities in body plan formation in animals, demonstrating that this complex plays a key role in this process [35]. The repressive activity of PRC2 is increased when the complex is accessorized with PALI1 [33]. It is thought that due to a shared domain between PALI1 and PALI2, PALI2 can likely interact with PRC2 in the same way; though as of the time of writing, no functional data have been published for PALI2.

It is presently unclear what molecular variation is responsible for the influence this region has on animal growth. Thus, the objective of this study was to comprehensively define the extended haplotype associated with increased growth by determining all potential variants that can be considered in-phase or not-in-phase with the haplotype using 34 Charolais-sired calves, 17 of which are *QQ* (homozygous for bearing the mutant haplotype associated with increased growth) and 17 which are *qq* (homozygous for ancestral haplotypes; or in other words not bearing the *Q* haplotype). These variants were then used to explore the associated effects on phenotype by genotyping a larger, multi-breed population for which several growth and carcass traits had been collected.

## 2. Materials and Methods

### 2.1. Sample Collection and Sequencing

Whole-blood samples were used from 439 cattle across six contemporary groups conducted at the University of Illinois in accordance with the animal care and use committee protocol associated with this project (IACUC Protocol #19118). DNA was extracted from these samples using a classic salting-out procedure [36] for populations 0, 1, 2, and 3. For populations 4 and 5, DNA was extracted using a Quick-DNA kit (Zymo Research, Irvine, CA, USA) following the manufacturer’s protocol.

For the initial objective, 34 Charolais-sired calves were selected for whole-genome sequencing. These animals were previously shown to be homozygous for the *QQ* (*n* = 17) and *qq* (*n* = 17) haplotypes within the region of interest [30]. DNA libraries were prepared using an Illumina DNA Prep library kit (Illumina, San Diego, CA, USA), following the manufacturer’s instructions. Libraries were paired-end sequenced using both lanes of a Novaseq 6000 S2 flowcell at 2 × 150 cycles.

Read quality was assessed using FastQC version 0.11.9 [37], and trimming was carried out using Trimmomatic version 0.39 [38] in paired-end mode using the following parameters: HEADCROP:1 ILLUMINACLIP:2:30:10 LEADING:28 SLIDINGWINDOW:15:28 MINLEN:75. Reads were aligned to the ARS-UCD 2.0 assembly of the bovine genome using BWA-MEM [39] with default settings. Alignments with a MAPQ below 30 were filtered out to remove poor-quality mappings and multi-mapped reads, before being passed to GATK 4.2.2.0 for final processing and variant calling [40]. Duplicates were flagged using MarkDuplicatesSpark and variants were called using HaplotypeCaller, both using default parameters. The variant call files were merged with GenomicsDBImport and joint genotyped with GenotypeGVCFs to create a combined VCF with all animal genotypes. Lastly, variants were hard-filtered with VariantFiltration with the following filters: “QD < 5.0”, “SOR > 2.5”, -filter “FS > 20.0”, and “MQ < 50.0”. Mapping Quality Rank Sum (MQRS) and Read Position Rank Sum (ReadPosRankSum) were not used due to their inability to be calculated when there are no samples heterozygous for a variant present. The choice of using *QQ* and *qq* animals for sequencing required that the variants of interest not be heterozygous, so filtering by those parameters would have removed them. For similar reasons, as well as for quality control, multiallelic variants were also removed.

### 2.2. Identification of Haplotype-Defining Mutations

To identify the variants that could be causing the *QQ* growth phenotype, a subtractive approach was taken. Using 17 *QQ* and 17 *qq* animals known from previous haplotype and phenotype information, the haplotype in this population was refined to a span between chr6:37,199,897–38,014,080, where almost all variants are entirely fixed in *QQ* individuals. To assess the validity of variants that did not present as fixed, they were visualized in Integrative Genomics Viewer (IGV) [41]. If called variants could be attributed to a single read or PCR duplicate or were found in an area where alignment might be impeded such as a structural variant or repetitive element, they were removed from the final list of variants for consideration. Filtering based on these criteria removed all but 5 variants that were not in phase with the *Q* haplotype. To determine variants in the haplotype that defined *Q*, the list was further narrowed to variants fixed in *QQ* animals, but where all *qq* individuals were homozygous for the opposite allele. Under the assumption that Dominette was *qq*, any variants where the *Q*-exclusive variant matched the reference were also removed. By filtering based on these criteria, the potential variant set was refined to variants exclusive to and completely in phase with the *Q* haplotype in this population.

Using Ensembl’s Variant Effect Predictor (VEP) [42], this set of variants was annotated based on location in the annotated genome (intronic, coding sequence, untranslated regions, intergenic, etc.), as well as impacts on coding sequence and reading frame. Effects on splicing were predicted with Pangolin [43] using the default settings. To investigate potential changes in transcription factor binding, variants were loaded into the UCSC genome browser as a custom track [44], alongside the bovine Assay for Transposase Accessible Chromatin using the sequencing (ATAC-seq) peak catalog published by Yuan et al. [45]. Variants located within ATAC-seq peaks and thus potentially affecting transcription factor (TF) binding were identified by visualizing the data in the browser. For further analysis, only variants within peaks reaching a signal score of at least 0.5 were considered. The effects of those variants were further interrogated using the Transcription Factor Binding Site Prediction tool provided by AnimalTFDB4.0 [46] to identify specific TFs that could bind to the affected regions. Find Individual Motif Occurrence (FIMO) [47] was then used to calculate scores for the original and mutated versions of the sequence, using all motifs associated with that transcription factor from CIS-BP [48]. For the final comparison, the motif with the highest FIMO score possible was selected for each TF.

### 2.3. Individual Variant Genotyping

To validate the presence of these variants in a broader population and to assess their potential as surrogates for the haplotype and predictors of phenotype, the remaining 405 animals were genotyped for two coding sequence variants found in *LCORL* exclusive to the *Q* haplotype, rs109696064 (chr6:g.37403795T) and rs384548488 (chr6:g.37401771_37401772del). For populations 1, 2, and 3, genotypes were called using a PCR-RFLP assay. Primers and enzymes used are described in Appendix A. PCR was conducted in 20 µL reactions containing 100 ng genomic DNA, 0.5U HotStarTaq DNA polymerase (QIAGEN, Valencia, CA, USA), 1× PCR buffer, 200 µM of each dNTP, and 0.5 µM each of forward and reverse primers. Amplification was performed with an initial incubation at 95 °C for 3 min, followed by 34 cycles of 94 °C for 20 s, 45 s, and 72 °C for 45 s, with a final incubation at 72 °C for 5 m. Following PCR, 10 µL of master mix (1.5 µL of 10× enzyme buffer, 5 Units of restriction enzyme, and 8 µL nuclease-free water) was added to each PCR reaction and incubated for one hour at 37 °C. The digested fragments were subjected to electrophoresis in 1.5% agarose, 0.5× tris-borate-EDTA gels with 0.1 µg/mL ethidium bromide. Genotypes were visualized by UV illumination. For populations 4 and 5, a fluorescent 5′–3′ exonuclease assay was performed using primers and probes as described in Appendix A. Quantitative PCR was performed in 10 µL reactions containing 1× PrimeTime™ Gene Expression Master Mix (Integrated DNA Technologies, Coralville, IA, USA), 0.5 µM each of forward and reverse primers, and 0.2 µM of each allele-specific probe. Amplification was performed with an initial incubation at 95 °C for 3 min followed by 40 cycles of 95 °C for 15 s, 60 °C for 45 s. Genotypes were called using CFX Maestro software 2.3, version 5.3.022 (Bio-Rad Laboratories, Hercules, CA, USA).

### 2.4. Statistical Analysis

Statistical analyses were performed in R version 4.3.2. lmer() [49] was used to construct linear mixed-effect models using genotype and sex as a fixed effect and farm as a random variable. Due to a lack of individuals bearing the alleles used as surrogates for the *QQ* genotype, populations 1 and 3 were not used for analysis. The final models for the first set were constructed using populations 2, 4, and 5 combined. Due to the difference in background genetics in these populations, population was used as a random effect to account for potential epistatic effects arising from those differences, as well as other differences that could not be accounted for between population groups, such as environment. Genotype and sex were used as fixed effects. Association was tested between genotype and 13 phenotypes: birth weight (BW), adjusted weaning weight (WW), three weight points through life (W1, W2, and W3), average daily gain (ADG), hip height (HH), dry matter intake (DMI), hot carcass weight (HCW), backfat thickness (BF), ribeye area (REA), kidney pelvic heart fat (KPH), and marbling (MB). As of the writing of this manuscript, DMI and carcass phenotypes were not available for populations 4 and 5, so only population 2 was considered for DMI, HCW, BF, REA, KPH, and MB. Additionally, only steers had HH and W3 measured in populations 4 and 5, although there were measurements from steers and heifers for these traits in population 2. Due to being calved considerably later than their contemporaries, five animals from population 4, and 34 from population 5, were not used for analyses. To adjust for multiple testing, the Benjamini–Hochberg correction was used [50]. All phenotypes within classes broken down by population and sex passed the Shapiro–Wilks normalcy test (*p* > 0.05), with the exception of W2, ADG, and MB for the steers of population 2, and birth weight for the steers of populations 4 and 5.

## 3. Results

### 3.1. Defining Mutations Exclusive to the QQ Haplotype

Thirty-four Charolais calves were whole-genome sequenced (17 *QQ* and 17 *qq*) with the goal of building a complete list of the potential causative mutations to better understand the functional impact of this haplotype. An average of 119.1 million read pairs were generated per sample, with 103.3 million read pairs remaining after trimming. After the removal of reads with a mapping quality below 30, each sample had 170.7 million reads mapped to the genome on average, resulting in a final coverage of roughly 9.2× for each sample, with the range of coverage for individual samples being between 7.6× and 11.5×.

As the ultimate objective was to investigate the *Q* haplotype at the *LCORL-NCAPG* locus, the first step was to identify the span of LD within this population. Based on the evidence from previous studies [9,30], the initial area of exploration encompassed 37,000,000 and 38,200,000 on BTA6, which contained *LAP3*, *MED28*, *FAM184B*, *DCAF16*, *NCAPG*, *LCORL*, and the 600 kb intergenic region upstream of *LCORL*. Within this area, an 814 kb region between 37,199,897–38,014,080 was shown to be in complete LD among the *QQ* animals, removing *LAP3*, *MED28*, and a portion of the 3′ end of *FAM184B* from consideration. In total, 7278 variants were called in this 814 kb region, 147 of which did not present as fixed in the *Q* haplotype. To validate these mutations, they were visualized in IGV. Of these variant calls, 81 could be attributed to a single read or PCR duplicate. Another 35 were in repetitive regions and could be construed as issues with alignment. There were three regions containing 21 total variants that were the result of failed alignment of larger repetitive regions. Five variants were clustered around apparent structural variants that are in phase with the haplotype, and the remaining five (rs109576691, rs379524098, chr6:g.37726299T>A, rs109270787, and rs383633472) did not fit any of these criteria and could be true germline variants existing within the *Q* haplotype or somatic mutations within individual animals. None of these five variants presented as being due to recombination events; the *Q* haplotype continued uninterrupted on either side of the variant in the individuals where they did not fit the defined *Q* genotype. Because of this, these variants were not useful in refining the *Q* haplotype further, nor are likely to be causative.

After quality control pruning, a total of 7131 SNPs and indels were identified as being fixed among *QQ* animals in this region. To be considered a variant ‘defining’ the haplotype, a variant had to be completely absent in *qq* individuals. In other words, if *QQ* animals were homozygous for one allele, the *qq* cattle had to be homozygous for the opposite allele. There were 217 variants that met these criteria. These variants were subject to further investigation to identify potential causative mechanisms using three prediction methods: VEP, for coding sequence and effects on reading frame; Pangolin, to detect changes in splicing sites; and combining published ATAC-seq data for cattle with FIMO predictions of binding affinity for transcription factors obtained from AnimalTFDB.

Of the 217 variants, only 22 were within the coding sequence or an ATAC peak (Table 1). No variants had a notable impact on splicing; the highest increase in splice probability score calculated by Pangolin was 0.02, and no decreases in splice score were observed. Of these 22 variants, three were detected in coding sequence: rs109570900 (the *NCAPGc.1326T>G* seen in previous studies), rs3845484488, and rs109696064. SIFT scores calculated for the SNPs indicate that rs109696064 is tolerated (0.38), but rs109570900 is deleterious (0.01). The last coding sequence variant, rs3845484488, is a frameshift that results in a truncation of the long isoform of the LCORL protein, so it is likely to be impactful. The other 19 variants were found in 18 ATAC peaks. Notably, rs109114124 and rs109092727 were found in the promoter region directly upstream of *LCORL*. However, neither of the variants in this region were in the major promoter ATAC peak, but in smaller adjacent peaks. The variant rs109145748 was located in the promoter for *FAM184B*, but the other regions were distributed among intronic and intergenic regions. A summary of the ATAC peaks these variants are in can be found in Table 2, and the differential transcription factor binding between the *Q* and *q* alleles for these variants is shown in Table 3.

While four of the variants in ATAC peaks had no predicted change in transcription factor affinity or any transcription factor affinity in their site, the remaining 15 had some degree of change. Most of the ATAC peaks harboring variants were relatively small, with only six having a signal score greater than 0.5. Notably for the variants near the *LCORL* promoter, although rs109092727 did not have any motif hits, the *Q* allele for rs109114124 results in a loss of affinity for EGR1 and EGR2 among others, but simultaneously a gain in affinity for FOXA1. A variant located in an embryonic ATAC peak, rs110458346, was also remarkable for its mutation causing a loss of affinity for all TFs predicted for that region in *qq*, including thyroid hormone receptors THRA, THRB, and SRF.

### 3.2. Discovery of Structural Variants

As part of quality control, regions with ambiguous calls were visualized in IGV. This led to the discovery of three apparent structural variants: (1) a 157 bp deletion within the first intron of *NCAPG* (chr6:37,326,536–37,326,693) (Appendix A), (2) an insertion within the fifth intron of *NCAPG* (chr6:37,336,715–37,336,716) (Appendix A), and (3) a small insertion in the intergenic space upstream of *LCORL* (chr6:37,619,145–37,619,149). As all three of these were within the fixed haplotype region for *Q*, they were homozygous in all *QQ* samples. Two of these structural variants were found to be present in *qq* samples. The smaller insertion upstream of LCORL was very common; 10 of the *qq* individuals were homozygous for this insertion and three were heterozygous. While rarer, the deletion in intron 1 of *NCAPG* was also present in *qq* animals, with four *qq* animals being heterozygous. However, the insertion within *NCAPG* seems to be completely absent among *qq* animals, and thus would qualify as being a defining mutation. This structural variant is displayed alongside all genotypes for the 1.2 Mb region in Figure 1. Mate pairs of reads entering the insertion point at this region appear on multiple different chromosomes, suggesting the insertion may be a repetitive element. The insertion is large enough that no mate-pair reads appear on the opposite side of the insertion point. This makes the insertion challenging to reconstruct using short reads alone, and it is unclear at this time if this insertion would have any impact on splicing or transcription of NCAPG.

### 3.3. Genotype–Phenotype Relationship

To confirm the existence of some of these mutations in a broader population, and to assess the association between selected SNPs and phenotype, an additional 405 cattle from five populations were genotyped for the rs384548488 and rs109696064 variants. These variants were selected due to their location in the coding sequence, the potentially significant impact of rs384548488, and to assess if either of these two very closely neighboring variants were found independent of one another. The variants were found to be in complete LD, with all animals being either homozygous ACT-C/ACT-C (as in ancestral haplotypes, or *qq*), A-T/A-T (*QQ*), or heterozygous for both (ACT-C/A-T, or *Qq*). Genotype distribution by population is listed in Table 4. Due to the proximity of these variants, and that these were the only variants genotyped, it cannot be confirmed if these cattle truly have the extended *Q* haplotype. However, given the evidence presented in the whole-genome sequenced animals, it may be possible to use these variants as a surrogate for the haplotype.

To assess the association between genotype and phenotype, linear mixed-effects models were constructed. As there were no *QQ* individuals in populations 1 and 3, only populations 2, 4, and 5 were used. Due to unavailability of DMI and carcass phenotypes for populations 4 and 5, only population 2 was used for the DMI, HCW, REA, BF, KPH, and MB phenotypes. The model constructed for each phenotype is presented in Table 5. BW, W2, W3, HH, ADG, and HCW all passed the significance threshold, with *Qq* and *QQ* cattle trending toward higher body weight, increased stature, and greater average daily gain than *qq* animals. This is consistent with the known effect of this locus on phenotype and demonstrates that these variants can serve as effective markers for the haplotype.

## 4. Discussion

The results of this investigation support the findings of numerous prior studies demonstrating the link between the NCAPG-LCORL locus and increased lean growth, and several variants have been identified that could underlie the changes in phenotype. In agreement with the results presented by Bouwman et al. [1] and others [9,13,25,51], there is a large region of LD in sequenced *QQ* animals, which unfortunately was not narrowed further compared to previous studies and continues to confound attempts to find a causative variant. Nevertheless, the list has been refined further by identifying variants exclusive to the haplotype and several potential molecular explanations for how this locus exerts its influence.

The most straightforward and tempting answers lie within the coding sequence mutations. The *NCAPGc.1326T>G* substitution, rs109570900, has been previously identified as a putative quantitative trait nucleotide (QTN) because the resulting missense substitution is predicted to be functionally damaging. Indeed, the *NCAPGc.1326G* allele was present in the 34 cattle sequenced here and was exclusive to the *Q* haplotype. Many studies investigating this region have found this SNP to be significantly associated with various phenotypes [2,9,31], and without doubt, it is in LD with this haplotype. However, other studies have questioned whether this mutation is causative. At least two studies have documented animals that would be heterozygous for the growth haplotype at this locus (*Qq*) being homozygous for the G allele, or animals that should not have the haplotype (*qq*) carrying a copy of this allele [29,32]. While none of the *qq* animals used in this study carried the G allele, it seems likely that this variant may exist outside of this haplotype in the broader population and may simply have been present on the chromosome on which the causative mutation first arose.

While evidence from previous studies suggests that rs109570900 is most likely not the causative variant, *NCAPG* itself cannot be fully ruled out, as indeed, expression of *NCAPG* seems to be important for muscle development. A recent study has found myogenic differentiation to be impaired in fetal myoblasts where *NCAPG* has been knocked down [52]. None of the 217 variants exclusive to *Q* were located within ATAC peaks in or directly upstream of *NCAPG*. However, the existence of an insertion within an intron of *NCAPG*, 84 bp downstream from an exon–intron junction, could potentially impact the transcription, splicing, or function of this gene, although more direct evidence will be necessary to confirm this impact.

Alternatively, the frameshift mutation of the long isoform of *LCORL* is quite compelling. The fact that other studies have documented a loss of function for this long isoform to be linked to increased stature in dogs and goats implies a similar mechanism could be at work here [18,26]. The frameshift variant in cattle, rs384548488, has been noted within the last year by Sanchez et al., Gualdrón Duarte et al., and Cai et al. to be associated with several beef production traits in mostly Charolais, Brown Swiss, and original Braunvieh cattle, height and length in Belgian Blues, and reduced young stock survival in Nordic Red cattle, respectively [53,54,55].

While the association with growth and carcass traits is unsurprising, given the body of literature surrounding this locus at this point, the influence on young stock survival observed by Cai et al. [54] is perhaps less expected. It may be easy to rationalize this effect as due to dystocia caused by the increase in birth weight associated with this genotype. Indeed, when considering stillbirths, almost the exact same region found in the current study’s results (chr6:37,236,226–38,027,078) was the most highly associated with the trait. The immediately adjacent proximal region (chr6:36,679,547–37,179,665) was the most significant region for calf survival within their first year, but the previously mentioned haplotype region (chr6: 37,236,226–38,027,078) still exceeded the significance threshold as well. It is curious that a mutation mostly known for its effect on growth is linked to early death, even after the first month of life. This locus was also linked to a decrease in longevity in the same study that identified the frameshift mutation in dogs [18]. That connection could simply be accounted for by correlation over causation with large breeds within a species tending towards a shorter lifespan, but it may also be that epigenetic changes spurred by the loss of function of PALI2, while also resulting in increased body frame, can impact longevity and survival. In their research on the epigenetic clock in dogs and humans, Horvath et al. [56] found regions that gained methylation with age were enriched for PRC2 targets and genes involved in development. Under the hypothesis that the long isoform of *LCORL* modulates PRC2 activity, there is an inviting connection to be made. However, much more evidence, particularly regarding epigenetic changes associated with this frameshift mutation, would be needed to draw this conclusion.

In a similar vein to the young stock survival locus being adjacent to, but not directly within the *LCORL-NCAPG* haplotype, Sanchez et al. [53] acknowledge that the frameshift mutation is not in very high LD with the lead SNP in Belgian Blues. Given the extensive LD in this region and the nature of GWAS for quantitative traits, caution should be exercised when looking at a lead SNP, as it may exist in a small number of higher-performing *Qq*/*qq* animals in addition to being attached to the *Q* haplotype. The fact that this region is in such LD is strong evidence for a selective sweep, and that the causative mutation is likely exclusive to the haplotype. Confirmation of further meiotic events to narrow down the haplotype further would be ideal. It is not lost on the author that over a decade ago, Setoguchi et al. suggested a considerably smaller, 591 kb haplotype region in Japanese Black cattle [9]. Translated locations of their markers to ARS-UCD2.0 would place their range at chr6:37,278,524–37,869,348, pruning roughly 144 kb from the region furthest upstream of *LCORL*. This recombination can be traced to sire C in their study. While the methods and reference genome then were different from today, it would be interesting to see how the haplotype presents in Japanese Black cattle, or if other recombination events can be found.

Setting aside the coding sequence mutations, the variants located in ATAC peaks, while perhaps not as straightforward an answer, are worthy of consideration as well. It has been shown previously that changes in *LCORL* expression have been linked both to this haplotype [30], as well as to feed intake [57]. Thus, changes in the regulation of *LCORL* may be contributing to this phenotype as well. Overall, changes to transcription factor binding trended toward loss of TF affinity in the *Q* haplotype; there are 37 TFs exclusive to *qq*, 22 exclusive to *QQ*, and 46 shared between the two haplotypes.

Though the distal region around chr6:37,900,000 would be excluded by the findings from Setoguchi et al., this region is fixed in the *Q* haplotype in the population genotyped in the present study. As this Charolais population is the same as those used in a study by Martins Rodrigues [30] to demonstrate increased *LCORL* expression in *QQ* individuals, it may still be worth considering the effects on transcription from this region. The site around rs110458346 is of particular interest, as it showed some of the strongest changes in TF affinity, and all resulted in a predicted loss of TF binding (Appendix A) in *QQ*. Additionally, this region is among the ATAC sites with a stronger signal and also is an embryonic-exclusive peak, suggesting a potential role in early developmental regulation. The most impacted transcription factors among these were the THRA and THRB thyroid hormone receptors. Thyroid hormone is important for normal embryonic development [58], and the presence of thyroid hormone receptor binding sites implies that the regulation of the genes at this locus may be thyroid-hormone-sensitive. However, the directionality of expression in response to T3 cannot be inferred from the sequence alone, as these receptors can promote or repress transcription in the presence of T3, depending on the other proteins involved in the complex at the locus [59]. The TF with the highest affinity for this site, SRF, is known for its important role in both development and skeletal muscle accretion [60,61], but similarly has fairly complicated and nuanced activity. SRF is most commonly known for promoting the expression of its target genes in response to growth factor stimulation [62], but it also can have a repressive action in competing with other transcription factors for binding sites [63].

In the case of the site around rs109114124, the most impactful changes to TF binding are the loss of affinity for the early growth response (EGR) transcription factors and the gain of FOXA1 binding. EGR1 and EGR2 are crucial regulators of cellular proliferation and apoptosis [64]. EGR1 has been demonstrated to activate pro-apoptotic and pro-survival pathways, again, depending on the context of the cell’s status as a whole [65]. FOXA1 is actually able to act as a ‘pioneer factor’, binding to condensed regions of chromatin and promoting the opening and transcription of previously inaccessible regions of the chromosome, although this is also dependent on the epigenetic marks of the histone as well as the sequence [66]. Thus, this gain of FOXA1 affinity interestingly suggests that individuals with the *QQ* haplotype may be able to promote *LCORL* transcription when it may otherwise be inaccessible for expression. Transcription factor activity ultimately relies on the coordinated activity of likely many cofactors interacting with each transcription factor associated with these binding sites, making it difficult to predict direct impacts caused by any individual variants by sequence alone. However, it does demonstrate that these changes likely affect the TF binding environment in some way and could warrant further investigation into these changes and their impact on transcription in the region and phenotype as a whole.

## 5. Conclusions

The *LCORL-NCAPG* locus is a critical region for growth in beef cattle, as evidenced by the findings presented here and the overwhelming body of evidence in the literature. This study has identified 218 mutations exclusive to the haplotype in the region associated with increased growth, including a structural variant in *NCAPG*, a frameshift variant causing a loss of function of the long isoform of *LCORL*, and several mutations affecting transcription factor binding and thus the potential regulation of genes in this locus. Genotyping for some of these variants showed statistically significant associations for birth weight, carcass weight, and average daily gain, making them useful markers for selection and prediction of performance. Though the true causative variant has yet to be determined due to the extensive LD in this region, these findings further clarify details underpinning this region and hopefully can contribute to future studies investigating how this locus mediates its effects.

## Figures and Tables

**Figure 1 genes-15-00576-f001:**
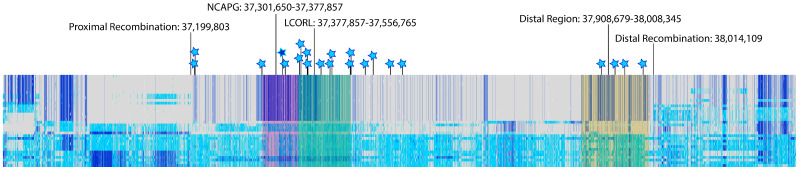
A display of all variants observed between chr6:37,000,000–38,200,000, with each horizontal bar being one of the genotyped cattle, and each vertical line representing the genotype for each variant. Gray represents homozygous for matching the reference (0/0), light blue is heterozygous (0/1), and dark blue is homozygous for the mutant allele (1/1). White indicates that a genotype was not able to be confidently called for that individual at that variant. *qq* animals are on the bottom, and the *QQ* haplotype is displayed on the top. Each star marks one of the 22 variants in phase with the haplotype that were found in the coding sequence or an ATAC peak, and the inverse-colored star marks where the structural variant in phase with the haplotype roughly is, though the SV itself, as well as the five variants not in phase with the haplotype, are not displayed on this map.

**Table 1 genes-15-00576-t001:** The 22 variants detected in the coding sequence or ATAC peaks.

Variant	Location ^1^	*q* Allele	*Q* Allele	Nearby Gene	Type, Consequence
rs109438687	37,214,389	T	C	*FAM184B*	ATAC peak, intron
rs109467519	37,214,736	C	T	*FAM184B*	ATAC peak, intron
rs109145748	37,301,160	G	C	*FAM184B*	ATAC peak, 5′ UTR
rs109570900	37,343,379	T	G	*NCAPG*	Coding sequence, missense
rs210386983	37,379,506	A	G	*LCORL*	ATAC peak, 3′ UTR
rs207496787	37,379,507	A	T	*LCORL*	ATAC peak, 3′ UTR
rs379449143	37,381,106	A	G	*LCORL*	ATAC peak, intron
rs384548488	37,401,770	ACT	A	*LCORL*	Coding sequence, frameshift
rs109696064	37,403,795	C	T	*LCORL*	Coding sequence, missense
rs517494305	37,452,882	C	CA	*LCORL*	ATAC peak, intron
rs110293947	37,479,269	G	C	*LCORL*	ATAC peak, intron
rs379787611	37,487,010	T	C	*LCORL*	ATAC peak, intron
rs109114124	37,555,677	C	A	*LCORL*	ATAC peak, intron
rs109092727	37,559,117	A	G	*LCORL*	ATAC peak, upstream
rs110470694	37,608,504	C	T		ATAC peak, intergenic
rs109060347	37,627,776	G	C		ATAC peak, intergenic
rs207689046	37,669,453	A	G		ATAC peak, intergenic
rs109331793	37,681,968	C	T		ATAC peak, intergenic
rs110458346	37,934,068	C	T		ATAC peak, intergenic
rs110888204	37,946,012	C	T		ATAC peak, intergenic
rs110930653	37,962,887	G	T		ATAC peak, intergenic
rs110658468	37,997,160	C	T		ATAC peak, intergenic

^1^ All variants are located on chromosome 6, ARS-UCD2.0 (NC_037333.1).

**Table 2 genes-15-00576-t002:** Consensus ATAC peaks with variants, their associated tissues, and signal scores.

Variant	Consensus Peak	Tissue	Signal Score ^1^
rs109438687	chr6_37214241_37214456_NMF12_0.33	Liver & Testicle	0.353
rs109467519	chr6_37214731_37214965_NMF10_0.92	Muscle	0.209
rs109145748	chr6_37300595_37301446_NMF10_0.14	Ubiquitous	0.909
rs210386983 & rs207496787	chr6_37379341_37379516_NMF13_1.00	8-cell Embryo	0.260
rs379449143	chr6_37381082_37381259_NMF13_1.00	8-cell Embryo	0.302
rs517494305	chr6_37452831_37452975_NMF16_0.67	Adipose	0.222
rs110293947	chr6_37479127_37479316_NMF7_0.58	Cerebellum	0.277
rs379787611	chr6_37487007_37487238_NMF13_1.00	8-cell Embryo	0.337
rs109114124	chr6_37555562_37555843_NMF9_0.15	Ubiquitous	0.522
rs109092727	chr6_37558579_37559199_NMF14_0.35	Embryo	0.866
rs110470694	chr6_37608344_37608531_NMF5_1.00	Colon & Embryo	0.422
rs109060347	chr6_37627598_37627866_NMF5_0.33	Colon, Rumen, Epithelial, & Embryo	0.947
rs207689046	chr6_37669337_37669559_NMF5_1.00	Colon	0.715
rs109331793	chr6_37681884_37682015_NMF13_1.00	8-cell Embryo	0.294
rs110458346	chr6_37933851_37934401_NMF13_1.00	8-cell Embryo	0.758
rs110888204	chr6_37945955_37946153_NMF10.88	Cerebrum	0.324
rs110930653	chr6_37962696_37962988_NMF16_0.49	Epididymis	0.198
rs110658468	chr6_37996798_37997347_NMF5_0.69	Colon	0.401

^1^ Signal score was determined by the highest signal in the entire consensus peak where the variant was found.

**Table 3 genes-15-00576-t003:** Transcription factors predicted to bind to the ATAC regions containing variants.

Variant	Shared TFs	*qq* TFs	*QQ* TFs
rs109438687	ZNF621	-	NR2C2, PAX6
rs109467519	-	-	-
rs109145748	GCM1, MAZ, SP2, ZNF180, ZNF212, ZNF341, ZNF467, ZNF527, ZNF548, ZNF596, ZNF792	PAX6, ZBTB14, ZFP64, ZNF264	KLF11, SP1, ZBTB17, ZNF329
rs210386983 & rs207496787	-	-	-
rs379449143	-	BATF3	POU6F1
rs517494305	NFATC1, RELA, ZNF484	NFATC3, REST	NFYA, NFYB, NFYC, ZNF280A, ZNF619
rs110293947	NFE2L2, NHLH1, NHLH2, OLIG2, TCF12, ZBTB18, ZNF273, ZNF331	ASCL2, MYOG, ZNF549, ZNF69, ZSCAN31	-
rs379787611	IRF1, STAT2, ZIM3, ZNF225, ZNF487, ZNF502	MEF2A, ZNF394	IRF2, IRF3, IRF4, IRF5, IRF8, IRF9, ZNF573
rs109114124	RREB1, ZNF263, ZNF283, ZNF785, ZNF805	EGR1, EGR2, MAZ, ZNF460, ZNF580	FOXA1
rs109092727	-	-	-
rs110470694	KLF15, ZNF383, ZNF432, ZNF880	-	ZNF449
rs109060347	ZNF335	NKX2-5	SOX18, ZNF200, ZNF808
rs207689046	ESR1, NR1H3, NR2C2, YY1	CREB3L1, CREB3L2, RORA	RXRG
rs109331793	CUX1, CUX2	ZNF667	ZNF605
rs110458346	-	MEF2B, POU6F1, SRF, THRA, THRB, ZNF774, ZNF823	-
rs110888204	ZNF768	PPARG, ZBTB12, ZNF543, ZNF621, ZNF768	ZNF440
rs110930653	-	NR1I2	-
rs110658468	-	-	-

**Table 4 genes-15-00576-t004:** Genotype frequencies by population.

Population	Breed Composition	*n*	*qq*	*Qq*	*QQ*
1	Simmental-Angus & Angus	30	18	12	0
2	Shorthorn	87	49	31	7
3	Angus	83	63	20	0
4	Simmental & Simmental-Angus	78	14	47	17
5	Simmental-Angus	127	62	59	6

**Table 5 genes-15-00576-t005:** Parameter estimates and statistical significance of genotype for each phenotype.

Phenotype	*n*	Intercept ^1^	β_*Qq*	β_*QQ*	β_Steer	Pop2	Pop4	Pop5	ASE ^2^	*p*-Value ^3^
Birth Weight (BW), kg	253	31.7 ± 2.3	0.4 ± 0.7	4.2 ± 1.1	3.1 ± 0.6	+3.5	−3.9	+0.5	1.6 ± 0.5	3.67 × 10^−4^ (*)
Adjusted Weaning Weight (WW), kg	241	206.8 ± 17.4	5.3 ± 4.6	9.5 ± 7.2	13.9 ± 4.1	+5.6	−31.6	+26.0	4.9 ± 3.3	0.327
Weight 1 (W1), kg	247	161.3 ± 54.5	7.0 ± 2.9	3.7 ± 4.7	12.5 ± 2.7	+108.9	−51.1	−57.8	3.5 ± 2.2	0.0603
Weight 2 (W2), kg	242	476.4 ± 12.9	22.6 ± 5.9	35.6 ± 9.7	56.0 ± 5.5	+20.7	−19.3	−1.4	19.4 ± 4.4	4.10 × 10^−5^ (*)
Weight 3 (W3), kg	157	517.8 ± 21.4	31.0 ± 7.7	32.2 ± 13.0	55.3 ± 9.6	+37.3	−18.1	−19.2	21.7 ± 5.7	1.40 × 10^−4^ (*)
Average Daily Gain (ADG), kg	244	1.46 ± 0.12	0.10 ± 0.03	0.18 ± 0.04	0.37 ± 0.02	−0.21	+0.01	+0.20	0.09 ± 0.02	2.27 × 10^−5^ (*)
Hip Height (HH), cm	157	120.9 ± 1.3	1.8 ± 0.7	2.6 ± 1.2	3.4 ± 0.9	+1.6	−1.1	−0.6	1.5 ± 0.5	0.0185 (*)
Dry Matter Intake (DMI), kg	87	9.02 ± 0.15	0.31 ± 0.21	0.44 ± 0.36	0.47 ± 0.20	-	-	-	0.26 ± 0.15	0.221
Hot Carcass Weight (HCW), kg	83	350.2 ± 5.4	21.4 ± 7.4	16.7 ± 13.6	32.5 ± 7.0	-	-	-	14.2 ± 5.5	0.0152 (*)
Backfat (BF), cm	83	1.59 ± 0.06	0.04 ± 0.09	−0.30 ± 0.16	−0.20 ± 0.08	-	-	-	−0.07 ± 0.07	0.130
Ribeye Area (REA), cm^2^	83	82.4 ± 1.5	5.2 ± 2.0	2.8 ± 3.8	7.0 ± 1.9	-	-	-	3.1 ± 1.5	0.0423
Kidney Pelvic Heart Fat (KPH), %	83	2.11 ± 0.03	−0.06 ± 0.04	−0.15 ± 0.08	−0.20 ± 0.04	-	-	-	−0.07 ± 0.03	0.102
Marbling (MB)	83	538.5 ± 15.5	32.1 ± 21.2	−12.5 ± 39.0	−73.3 ± 20.2	-	-	-	10.9 ± 15.8	0.267

^1^ Intercept for these models is *qq* heifer, with β from *Qq*, *QQ*, and/or sex effect for steers being added as applicable. Population is included as a random effect added to the intercept. For DMI, HCW, BF, REA, KPH, and MB, there is no farm effect, due to only population 2 being phenotyped for these traits. ^2^ Allele substitution effect (ASE) was calculated using haplotype as an additive instead of a categorical variable. There was no change in which traits passed the significance threshold between using a categorical or additive model. ^3^ Calculated using haplotype as a categorical variable. Models where genotype exceeded the Benjamini–Hochberg adjusted-significance threshold are denoted by (*).

## Data Availability

Raw fastqs from all animals will be available on NCBI SRA.

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
