# Peer review of "Defining a Haplotype Encompassing the LCORL-NCAPG Locus Associated with Increased Lean Growth in Beef Cattle"

_genes, 2024, doi:10.3390/genes15050576_

Round 1
Reviewer 1 Report
Comments and Suggestions for Authors
The LCORL-NCAPG locus is known for its association with growth traits in cattle. In this study, evidence from this study underscores its significance, having identified more than 200 variants, including variants in LCORL and NCAPG genes, respectively. Moreover, the authors detected candidate causal variants in the transcriptional regulatory elements in this locus. Selected variants were associated with birth weight, carcass weight, and growth rate, highlighting their potential as markers for MAS or genomic selection. The manuscript was well-written and readable. However, I would like to make several comments that should be addressed.
1. The frequencies of all the haplotypes estimated by HaplotypeCaller should be present as a table.
2. Please add the LD heat map plot for chr6:37,000,000-38,200,000 region under the current panel of Figure1.
3. In addition to beta_Qq and beta_QQ columns, please add one more column for the additive effect (i.e., allelic substitution effect) with its standard error.
4. Please compute percent variation explained by this LCORL-NCAPG locus as the effect size of this locus on the phenotype of interests.
5. Please calculate an ad hoc statistical power study for each significant association and discuss the impact of the power study on this study in general.
6. In this study, author constructed the mixed linear model for the association study including population as random, please explain why did they designate the population as random effect and discuss it with the aspect of correcting population stratification.
Reviewer 2 Report
Comments and Suggestions for Authors
Dear Authors,
the work presented by you about LOCORL-NCAPG locus, critical for growth in beef cattle is methodologically very broad with different and relevant obtained outcomes, as correctly elucidated in discussion section.
But in general the manuscript, including the adopted approaches and the results are not clearly presented and/or valorised, starting from the abstract.
For example, paragraph at lines 63 and following lines is confusing for readers. In Materials and Methods paragraphs and sub-paragraphs are lacking, generating a lot of confusion.
The aim of the work is "lost" in misleading concepts, so please emphasize it better framing better also the topic.
The paragraph at lines 229-234 should be moved in materials and methods section.
Please, specify acronyms IGV, ATAC,....
Is the q haplotype the wild-type allele and the Q haplotype the mutated one? In some sections of the manuscript, it is not clear or obvious, starting from Table 1.
The paragraph at lines 276-284 is not clear at all.
The resolution of Figure 1 is not optimal.
Best regards.
Comments on the Quality of English LanguageModerate editing of English language required
Reviewer 3 Report
Comments and Suggestions for Authors
The researchers used 34 whole genome sequences (~9.2x) of Charolais calves. Half of them are QQ (increased growth) and the other half are qq (ancestral). They found 218 variants in an 814-kb region that were not in qq individuals and annotated them in gene and ATAC regions. The genotypes' two coding sequence variants of for 405 cattle were used for their linkage analysis. Finally, they found significant associations between genotypes and birth weight, live weight, carcass weight, hip height, and average daily gain. This paper is interesting, and well designed. Three recommends listed below:
“3.2. Discovery of Structural Variants”:
Structural variants refer to large-scale genomic alterations, such as insertions, deletions, duplications, inversions, and translocations. These variations can impact gene function and contribute to phenotypic diversity. In the methods section, the authors should clarify how they identified structural variants. Did they use specialized algorithms, comparative genomics, or other techniques? Providing this information will enhance the paper’s clarity.
“3.3. Genotype-Phenotype Relationship”:
Linear mixed-effects models (LMMs) are statistical tools used to analyze relationships between variables while accounting for random effects (e.g., individual variability). In this study, LMMs were likely employed to assess how genotypes (QQ and qq) relate to phenotypic traits (birth weight, live weight, etc.). The cited paper by Bates et al. (2015) provides a comprehensive guide to fitting LMMs using the lme4 package in R. Authors should briefly describe their LMM approach, referencing this paper for readers seeking more details.
Add subtitles to the methods section.
Round 2
Reviewer 1 Report
Comments and Suggestions for Authors
The authors addressed my previous comments properly.
Reviewer 2 Report
Comments and Suggestions for Authors
The authors had clarified my doubts and they have amended the manuscript accordingly, even if I could not find the corrections tracked in the new version of the paper.
Best regards.
Comments on the Quality of English Language
Moderate editing of English language required